# Fabrications of Electrospun Mesoporous TiO_2_ Nanofibers with Various Amounts of PVP and Photocatalytic Properties on Methylene Blue (MB) Photodegradation

**DOI:** 10.3390/polym15010134

**Published:** 2022-12-28

**Authors:** Sun-Ho Yoo, Han-Sol Yoon, HyukSu Han, Kyeong-Han Na, Won-Youl Choi

**Affiliations:** 1Department of Advanced Materials Engineering, Gangneung-Wonju National University, 7 Jukheongil, Gangneung 25457, Republic of Korea; 2Department of Energy Engineering, Konkuk University, 120 Neungdong-ro, Gwangjin-gu, Seoul 05029, Republic of Korea; 3Research Institute for Dental Engineering, Gangneung-Wonju National University, Gangneung 25457, Republic of Korea; 4Smart Hydrogen Energy Center, Gangneung-Wonju National University, 7 Jukheongil, Gangneung 25457, Republic of Korea

**Keywords:** porous TiO_2_, electrospinning, K-90 PVP, K-30 PVP, photocatalyst

## Abstract

The superior chemical and electrical properties of TiO_2_ are considered to be suitable material for various applications, such as photoelectrodes, photocatalysts, and semiconductor gas sensors; however, it is difficult to commercialize the applications due to their low photoelectric conversion efficiency. Various solutions have been suggested and among them, the increase of active sites through surface modification is one of the most studied methods. A porous nanostructure with a large specific surface area is an attractive solution to increasing active sites, and in the electrospinning process, mesoporous nanofibers can be obtained by controlling the composition of the precursor solution. This study successfully carried out surface modification of TiO_2_ nanofibers by mixing polyvinylpyrrolidone with different molecular weights and using diisopropyl azodicarboxylate (DIPA). The morphology and crystallographic properties of the TiO_2_ samples were analyzed using a field emission electron microscope and X-ray diffraction method. The specific surface area and pore properties of the nanofiber samples were compared using the Brunauer-Emmett-Teller method. The TiO_2_ nanofibers fabricated by the precursor with K-30 polyvinyl pyrrolidone and diisopropyl azodicarboxylate were more porous than the TiO_2_ nanofibers without them. The modified nanofibers with K-30 and DIPA had a photocatalytic efficiency of 150% compared to TiO_2_ nanofibers. Their X-ray diffraction patterns revealed anatase peaks. The average crystallite size of the modified nanofibers was calculated to be 6.27–9.27 nm, and the specific surface area was 23.5–27.4 m^2^/g, which was more than 150% larger than the 17.2 m^2^/g of ordinary TiO_2_ nanofibers.

## 1. Introduction

In the last decades, environmental pollution due to the increase in population and resource consumption has become a critical issue. Environmental pollution caused by industrial waste disrupts sustainable development and causes multiple problems. The use of fossil fuels, such as coal, oil, and natural gas, increases the amount of carbon dioxide and fine dust in the atmosphere, and causes air pollution [1,2,3,4,5,6,7]; in addition, various types of domestic and industrial wastewater cause water pollution [8,9]. Many research studies on the capture, storage, and removal of pollutants have been conducted and solutions proposed for sustainable development [10,11,12,13,14]. Among them, the degradation of pollutants using semiconductor photocatalysts is one of the important ideas [15,16,17,18,19,20,21]. A photocatalyst for the degradation of water pollutants requires various properties, such as non-toxicity, chemical resistance, and an electronic structure that can use sunlight; in addition, TiO_2_ has attracted much attention as a photocatalyst due to its superior chemical and electrical properties [22,23,24]. However, the industrial application of TiO_2_ has been limited because of its poor photoelectric conversion efficiency, which is due to its wide band gap (~3.2 eV) and the rapid recombination rate of electron-hole pairs [25,26,27,28,29]. Synthesis methods, such as hydrothermal, microwave, anodization, chemical vapor deposition, and electrospinning, have been reported for the fabrication of various TiO_2_ nanostructures [30,31,32,33,34,35,36,37]; in addition, among them, electrospinning is a simple and economic method to obtain uniform and continuous one-dimensional nanofibers. Electrospinning is a process to obtain nanofibers with a nanoscale diameter through jet stretching of a viscous polymer solution by an electric field and rapid solvent volatilization. At the same time, the improvement of nanofibers, such as element doping and morphology modification, can be achieved simply by controlling the composition of the precursor solution [38,39,40,41,42,43,44,45]. In addition, to improve photocatalytic activity, studies on the synthesis of TiO_2_ nanofibers containing noble metals and transition metal dopants, such as Ag, Cu, C, Fe, and Co, have been prominently reported [46,47,48,49,50].

In this study, we modified mesoporous nanofibers using polyvinylpyrrolidone with different molecular weights and diisopropyl azodicarboxylate (DIPA) for a precursor solution. Previous studies using titanium butoxide, polyvinyl pyrrolidone (PVP), and polyvinyl alcohol have been reported for the mesoporous electrospun nanofibers fabrication; however, studies using a titanium tetraisopropoxide (TTIP) and PVP mixture of different molecular weights are difficult to find. The surface and pore properties of the nanofibers were improved via control of the calcination temperature ramping speed. To compare the modifed TiO_2_ nanofibers (TNF) and ordinary nanofibers, a methylene blue (MB) photodegradation test was carried out using a UV lamp, and the absorbance of the MB solution was measured using UV–Vis. The test results confirmed that the modified nanofibers achieved improved photocatalytic efficiency over ordinary nanofibers [51,52].

## 2. Materials and Methods

### 2.1. Materials

PVP (K-90, polyvinyl pyrrolidone, M.W. 1,300,000), (K-30, polyvinyl pyrrolidone, M.W. 58,000), and DIPA (diisopropyl azodicarboxylate, 94%) were purchased from Alfa Aesar Korea Co., Ltd. (Incheon, Republic of Korea). TTIP (titanium tetraisopropoxide, ≥98.0%) and ACAC (acetyl acetone, ≥99.0%) were purchased from Junsei Co., Ltd. (Tokyo, Japan). EtOH (ethyl alcohol, ≥99.5% EP) was purchased from Daejung (Siheung, Republic of Korea).

### 2.2. Fabrication of TiO_2_ Nanofibers

As shown in Table 1, precursor solutions were prepared under different compositions and calcination conditions. First, 10 g of PVP was added to EtOH 50 g and stirred for 24 h using a magnetic stirrer. TNF 1, 2, and 3 were prepared using 5 g of K-90 and 5g of K-30. After that, a second solution was prepared by mixing 15 g of TTIP and 10 g of ACAC in another beaker for 3 h. Then, the second solution was added to the first solution and stirred for 3 h; then, 10 g of DIPA was added to the mixture solution and vigorously stirred until a yellow transparent solution was obtained.

The prepared precursor solutions were electrospun using a nozzle capillary with a diameter of 0.34 mm (23 gauge) at each 1.0 mL/h flow rate. The tip-to-collector distance was kept at 15 cm, and the applied voltage was set to 15 kV. The process was carried out at room temperature and humidity below 50%, and aluminum foil was used as the collector.

A schematic diagram of the electrospinning process is shown in Figure 1. A 3-tip multi-nozzle was used in the process, and 4 h was set as 1 batch. The weight of as-spun nanofibers collected per batch was about 4 g; then, dried and calcined.

### 2.3. Characterization

As-spun nanofibers were dried in an oven heated to 60 °C for 2 h and then, calcined by heating in a box furnace to 500 °C.

Thermogravimetric analysis (TGA, Q500, TA Instruments, Newcastle, DE, USA) was carried out to decide the calcination temperature of as-spun nanofibers. The surface and average diameter of each sample were measured using a field emission scanning electron microscope (FE-SEM, Inspect F, FEI Korea Co., Ltd., Gyeonggi-do, Republic of Korea). FT-IR spectra of the calcined TiO_2_ nanofibers were obtained using a Fourier transform infrared spectrometer (FT-IR, iS50, Thermo Fisher Scientific, Waltham, MA, USA), with a KBr pellet in the range 400 and 4000 cm^−1^. The crystal structure and crystallinity of the TiO_2_ sample were analyzed using the diffraction pattern obtained using an X-ray diffractometer (XRD, AXS-D8, Bruker Korea Co., Ltd., Gyeonggi-do, Republic of Korea). The specific surface area and pore characteristics were analyzed using a Brunauer-Emmett-Teller surface area analyzer (TriStar II 3020, Micromeritics, Norcross, GA, USA), and the adsorption/desorption of N_2_ gas was carried out at 77.3 K.

### 2.4. Photocatalytic MB Degradation

The MB degradation test was carried out in a dark room to compare the photocatalytic efficiency of each sample. An aqueous solution of 5 mg/L was prepared using MB as a pollutant; in addition, 0.2 g of the TiO_2_ nanofiber sample and 20 mL of deionized water were added to a quartz beaker, and stirred for 30 min. After that, 200 mL of MB aqueous solution was added to the TiO_2_ dispersed water and stirred for 2 h in a dark room, with temperature control for mixing and stabilizing. After that, the distance between the UV lamp (6 W, 365 nm) and the quartz beaker was fixed at 10 cm, and the beaker was stirred at 240 rpm during the photodegradation reaction. The reaction was carried out for 3 h, and the mixed solution was sampled every 30 min using a syringe. The TiO_2_ photocatalyst in the sampled solution was filtered and removed using a syringe filter (PVDF filter, 0.2 μm, Whatman, Marlborough, MA, USA); in addition, the filtered solution was stored in a cuvette and refrigerated to block the incident light.

## 3. Results and Discussion

A TGA analysis was carried out to determine the calcination temperature of TNF, including a blowing agent. The thermal behavior of TNF0 and TNF1 was compared to confirm that the same calcination temperature of ordinary TNF could be applied, and the results are shown in Figure 2. After setting the initial temperature to 30 °C and stabilizing, it was heated to 500 °C for a heating rate of 5 °C/min. Both the balance gas and sample gas flow rates were set to 50.0 mL/min of N_2_ gas, and each TNF sample weighed 10 mg. The change of the TGA curve was observed in three steps, and the evaporation of the solvent and adsorbed water evaporation was not significant due to the drying process. The first section at ~150 °C was considered as mass reduction via the evaporation of moisture and residual solvents. Starting at 210 °C, the second section can be considered the glass transition and combustion phase. At this time, the decrease rate of TNF 1 was greater than that of TNF 0, which is considered due to the lower burning point of DIPA and low molecular weight PVP. When the section is above 390 °C, the thermal decomposition of the carbonized compound and the residual polymer starts. After 450 °C, the final weights of TNF 0 and TNF 1 were 29.6 wt% and 32.1 wt%, respectively; in addition, it was confirmed that there was no significant difference and the thermal behavior was ended. The calcination temperature was determined to be 500 °C to prevent rutile growth starting at 600 °C and higher, while providing a sufficient driving force for crystallization [53,54].

The surface structure and microstructure of the TiO_2_ nanofiber (TNF) samples were analyzed using FE-SEM as shown in Figure 3. Figure 3a,b are low and high magnification images of TNF 0, which are ordinary nanofibers fabricated via typical electrospinning. No morphological defects were observed on the nanofiber surface, and a uniform and smooth surface was observed. Figure 3c–h show low and high magnification images of TNF 1, 2, and 3 that used DIPA and a mixture of K-90 and K-30. Each sample was calcined for heating rates of 1 °C/min, 3 °C/min, and 5 °C/min. Compared to the TNF 0 sample, TNF 1, 2, and 3 showed a modified surface; and beads or bubbles were not observed. Depending on the heating rate during the calcination process, byproduct gases of thermal decomposition are emitted at different rates and induce surface deformation. In this experiment, as a chemical blowing agent, DIPA reacted with water to generate a large amount of CO_2_ and various gases. DIPA contained inside the nanofibers is gasified at high temperatures, and mesopores are formed where the gas was released. Since the speed at which the blowing agent is gasified and released depends on the heating rate, the pore properties can be controlled using this [55,56]. In particular, it was observed that TNF 3 had smaller pores compared to other samples.

More than 300 diameter values were measured using FE-SEM images, and the average diameter graph is shown in Figure 4. The average diameter values of TNF 1, 2, and 3 were verified to be 760 nm, 692 nm, and 507 nm, respectively. The value of TNF 0 was calculated to be 552 nm. Our results verified that the average diameter of calcinated nanofibers also decreased as the heating rate decreased. The results also verified that all of the modified nanofibers compared to TNF 0 had a larger average diameter, and this difference was related to the molecular weight of the polymer and the presence of a blowing agent. At the same time, the unstable nanofiber morphology and fracture of TNF1, 2, and 3 are also explained by the decrease in crystallinity and weakening of mechanical strength due to internal gas emission. Compared to dense and uniform TNF0, TNF1, 2, and 3, which have many defects in microstructure, they have mechanically weak properties [57,58]. The rheological behavior and relaxation of the precursor solution in electrospinning depend on the concentration of the polymer. The viscosity of precursors, including K-90, K-30, and DIPA, controls the surface tension when forming a Taylor cone at the nozzle during electrospinning, resulting in changes in the average diameter. It has been reported that the molecular weight of the polymer or other factors may interact with the diameter of the nanofibers [59,60].

FT-IR measurements were recorded using a tungsten halogen NIR lamp. Figure 5 shows the FT-IR spectroscopy of TiO_2_ nanofibers at different calcination heating rates. The analysis of chemical bonding and various compositions was performed by FT-IR in 4000 cm^−1^ and 400 cm^−1^, as shown in Figure 5. It can be seen that the various vibrational peaks in the observed spectrum are consistent with the reported literature [61,62,63,64]. In relation to the observed spectra of all the samples, the peaks are approximately presented at 3436, 2973, 2345, 1629, and 516 cm^−1^. The peak at 3432 cm^−1^ represents the O–H bonding stretching vibration associated with absorbed water. The peak at 2970 cm^−1^ is attributed to C–H stretching in the polymer (CH_2_). The peak is also due to the stretching of C=O found in PVP and has a wave number of 1650 cm^−1^. The vibrational peak at 516 cm^−1^ was found to be related to the characteristic Ti–O–Ti bonds.

XRD analysis was carried out to verify the crystal structure of the TNF samples and the results are shown in Figure 4. The XRD pattern was obtained from 20° to 80° under conditions of 0.02°/step and 1 step/s, using Co-kα radiation (λ = 1.789010 Å) and a 2theta method. 

Figure 6a shows the diffraction pattern of TNF 0, and the (110) and (200) planes of the anatase phase were confirmed at 29.4° and 56.5°, respectively. The peaks at 32.0° and 63.9° were identified as the (100) and (211) planes of rutile, respectively. As a result of Rietveld refinement to determine the ratio of the complex phase, the anatase was 80.8%, and that of rutile was 19.2%. TNF 1, 2, and 3 showed peaks at 29.4°, 44.5°, 56.5°, 63.9°, and 64.0°, which are indexed as (011), (004), (020), (015), and (121) planes, respectively, of a single anatase phase. The color of the as-spun nanofibers was yellow and then, changed to white after calcination. The average crystallite size of each sample was calculated using the Scherrer equation in (1) below.
D = K λ/ β cos θ(1)
where D represents the average crystallite diameter, K = shape factor, λ = wavelength of the wavelength of X-ray used for diffraction (1.789010 Å), β = full width at half maximum (FWHM) of the peak, and θ is the Bragg angle. The average crystallite sizes of TNF 0, TNF 1, TNF 2, and TNF 3 calculated by the Debye-Scherer formula were 31.98 nm, 6.27 nm, 6.67 nm, and 9.27 nm, respectively. According to the calculation, the crystallite size of TNF 1, 2, and 3 decreases as the heating rate increases.

The specific surface area and pore properties were confirmed using a BET method. Adsorption/desorption of N_2_ gas was carried out at 77.3 K and Figure 7 shows the obtained adsorption/desorption curve for N_2_. The specific surface area and pore properties calculated using BET theory and the Barrett-Joyner-Halenda (BJH) absorption/desorption method are summarized in Table 2. The obtained curve corresponds to types IV and V, among various types of the International Union of Pure Application Chemistry (IUPAC) classification. The hysteresis originated from the difference in adsorption/desorption partial pressure according to the substance. The shape of the adsorption/desorption curve for relative pressure means that the TiO_2_ nanofibers had a mesoporous surface with a large specific surface area. The specific surface area of each sample was calculated as 17.2 m^2^/g, 23.8 m^2^/g, 27.4 m^2^/g, and 23.5 m^2^/g for TNF 0, 1, 2, and 3, respectively. Pore volume means the volume of all the open pores formed on the surface of a material. The BJH desorption average pore width represents the average throat size of pores, and the micropore area represents the pore area per unit mass of the sample. Theoretically, an increase in the specific surface area leads to an increase in the active site and has a proportional relationship with the photocatalytic activity. However, in reality, the photocatalytic activity is rather reduced due to the surface tension of the solution, and the ease of adsorption and desorption of the generated gas, depending on the shape or diameter of the pores. These results show that the specific surface area of nanofibers can be increased by controlling the molecular weight of PVP and adding DIPA. Compared to TNF 0, the mesoporous nanofibers exhibited a specific surface area increased by up to 159%. The average pore widths of TNF 0, 1, 2, and 3 calculated by the BJH method were 13.67 nm, 15.64 nm, 8.7 nm, and 16.2 nm, respectively. TNF 2 was the highest in specific surface area, whereas TNF 3 had better characteristics, such as pore volume, size, and micropore area.

In order to compare the photocatalytic activity, we performed a methylene blue (MB) photodegradation test by TNFs (TiO_2_ nanofiber samples). Figure 8 shows the absorption spectrum of the MB aqueous solution sampled every 30 min during the photocatalytic reaction. In all of the TNF/MB solution mixtures, the intensity of the absorption peak decreased as time passed, which means that the MB molecules in the aqueous solution were decomposed by TNFs. Among them, the absorbance of TNF 1, 2, and 3 compared to TNF 0 was dramatically decreased, and it can be considered that the mesoporous surface properties positively contribute to the photocatalytic activity.

We obtained normalized values (C/C_0_) and ln (C_0_/C) by taking the absorbance at 665 nm to C value, where 665 nm is the lambda max of methylene blue. The C/C_0_ and ln(C/C_0_) values as time passed are shown in Figure 9. As seen in Figure 9, TNF 0 showed the lowest photolytic activity and excellent efficiency in the order of TNF 3, 2, and 1. When we considered only the specific surface area, this tendency differed from what we expected; however, it can be explained via pore properties [65]. In this case, since the active sites were relatively small, which can contribute to the photocatalytic reaction due to the surface tension of the aqueous solution and the absorbed byproducts on the surface, the photocatalytic activity decreased. In the case of TNF 1 and 3, the BET data were similar; however, there was a difference in the total pore area and crystallinity. Therefore, the antagonism of these properties in combination can be explained by the fact that the photocatalytic activity was high in the order of TNF 3, 2, 1. After UV irradiation for 3 h, the degradation rates of TNF 0, TNF 1, 2, and 3 were calculated to be 66.7%, 95.5%, 98.4%, and 99.7%, respectively. The aqueous solution sampled using a cuvette is shown in Figure 10. The sample in the cuvette is a solution in which a syringe filter filtered the catalyst, and the color degradation and transmittance can be observed with the naked eye.

## 4. Conclusions

To improve the photocatalytic activity of TiO_2_ nanofibers, we prepared modified surfaces using electrospinning and compared the photodegradation efficiency. The precursor solution composition for electrospinning was controlled using the PVP molecular weight and DIPA to obtain the modified nanofibers. An anatase single crystal phase, high specific surface area, and improved photocatalytic activity were achieved in the modified nanofibers. Fabricated nanofibers using DIPA and K-30 showed mesoporous adsorption/desorption curves, and the specific surface area increased from 17.2 m^2^/g to 27.4 m^2^/g compared to the ordinary nanofibers.

The photodegradation rate of methylene blue was increased up to 150%, which validates the improvement of the photocatalytic activity of modified nanofibers. Different crystallinity and surface properties were obtained according to the heating rate for the calcination and composition of precursor solutions; in addition, it was necessary to optimize these factors to improve the photocatalytic activity. To contribute to the potential of TiO_2_ for the field of photocatalyst materials, the mesoporous surface of TiO_2_ obtained using a simple method such as the one proposed here can be used with other techniques to improve the activity of photocatalysts, such as doping and heterointerface formation.

## Figures and Tables

**Figure 1 polymers-15-00134-f001:**
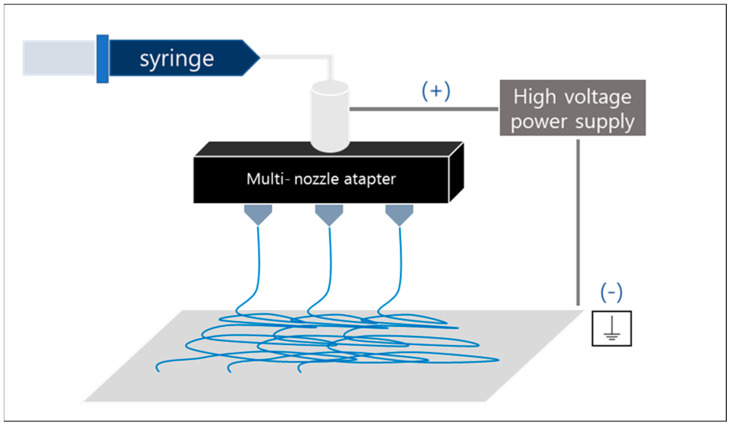
Schematic diagram of electrospinning process.

**Figure 2 polymers-15-00134-f002:**
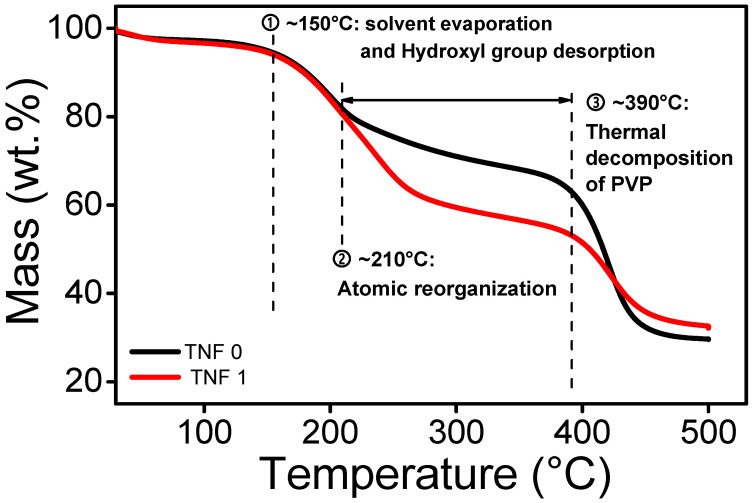
TGA curve of as-spun TiO_2_ nanofibers.

**Figure 3 polymers-15-00134-f003:**
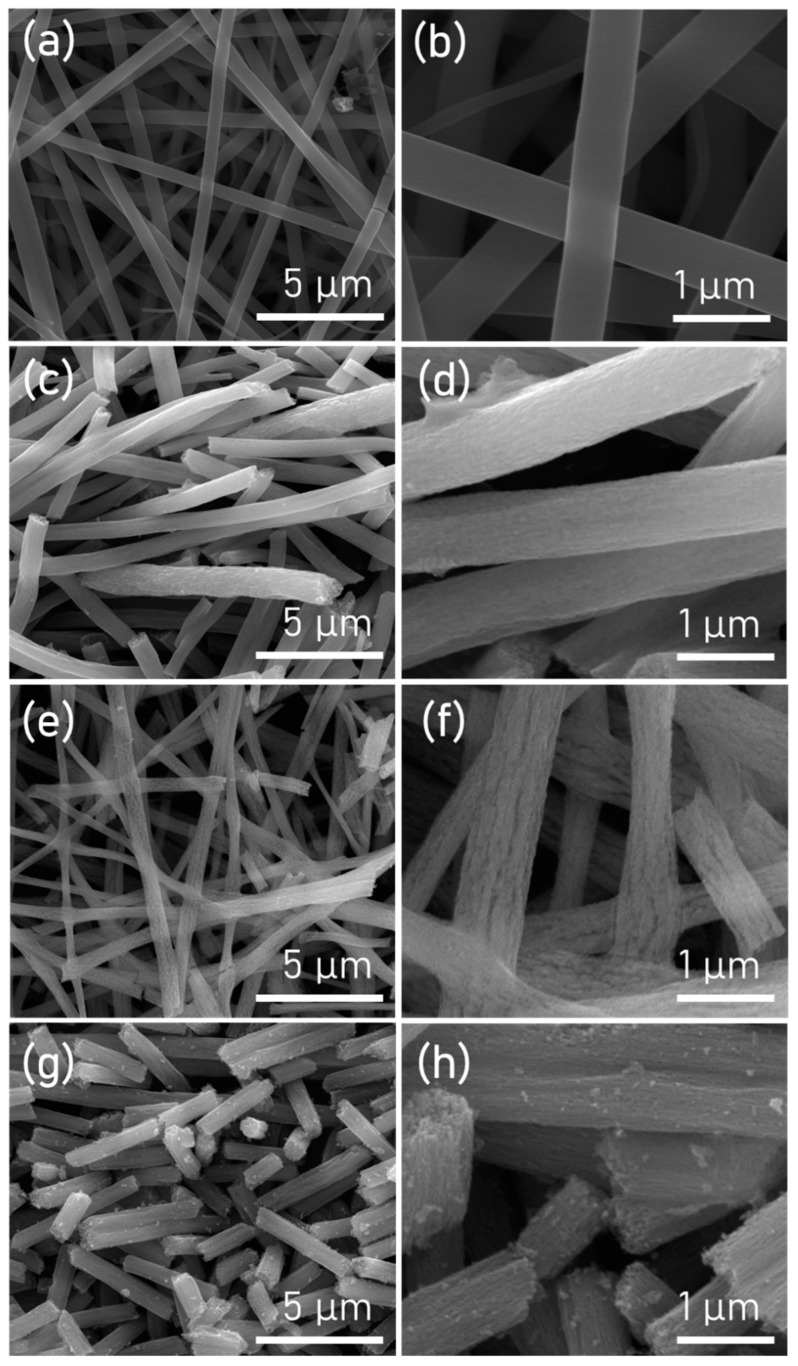
FE-SEM images of TNFs at low and high magnification: (**a**,**b**) TNF 0; (**c**,**d**) TNF 1; (**e**,**f**) TNF 2; (**g**,**h**) TNF 3.

**Figure 4 polymers-15-00134-f004:**
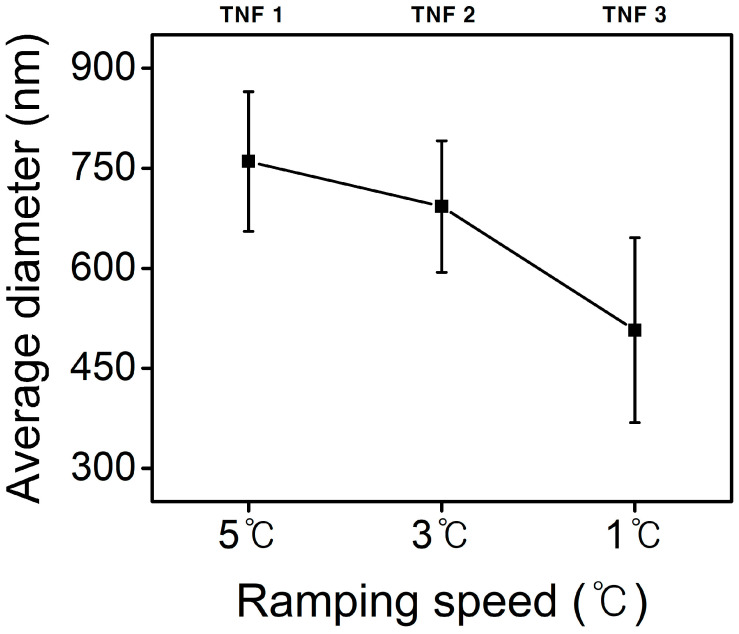
Average diameters of TNF samples with ramping speed.

**Figure 5 polymers-15-00134-f005:**
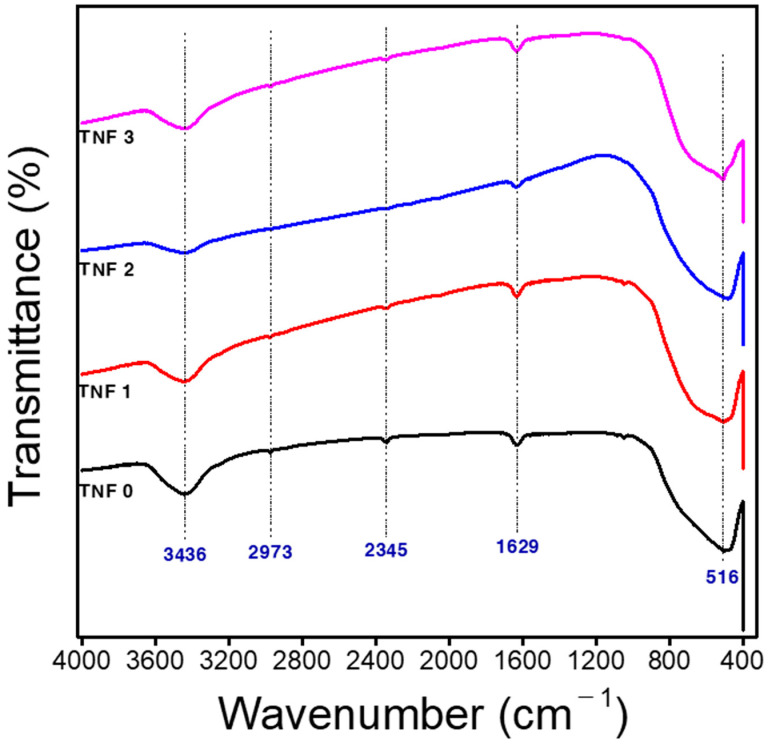
FT-IR spectra curves of electrospun TNF samples.

**Figure 6 polymers-15-00134-f006:**
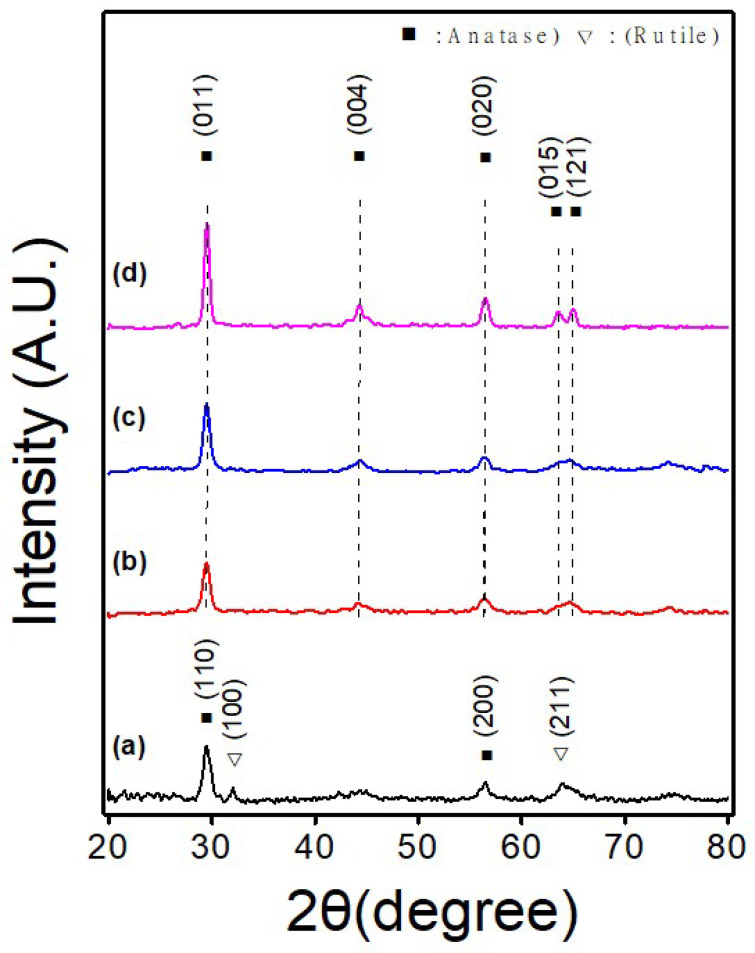
X-ray diffractometer (XRD) spectra pattern of various TiO_2_ nanofiber samples ((**a**): TNF 0; (**b**): TNF 1; (**c**): TNF 2; (**d**): TNF 3).

**Figure 7 polymers-15-00134-f007:**
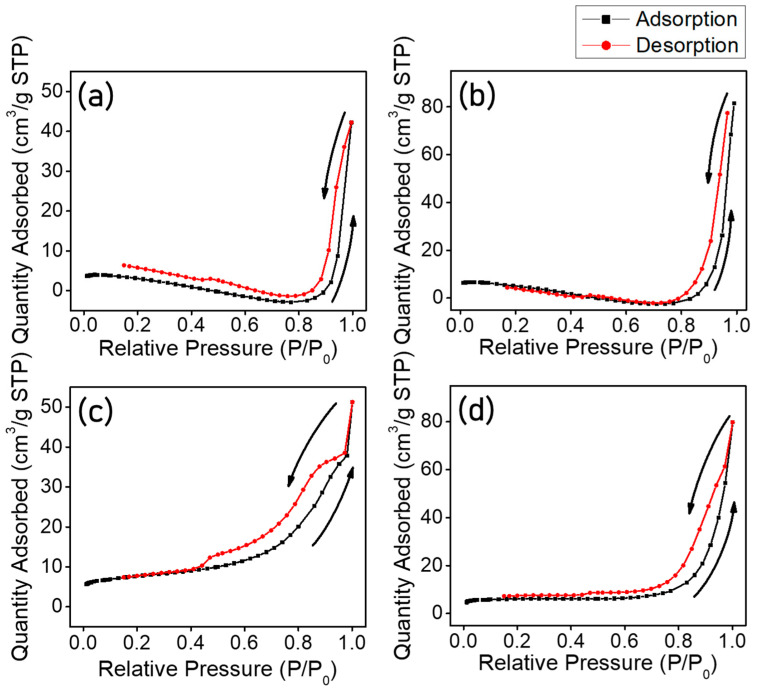
The isotherm of adsorption/desorption of N_2_ on the TNF samples ((**a**): TNF 0; (**b**): TNF 1; (**c**): TNF 2; (**d**): TNF 3).

**Figure 8 polymers-15-00134-f008:**
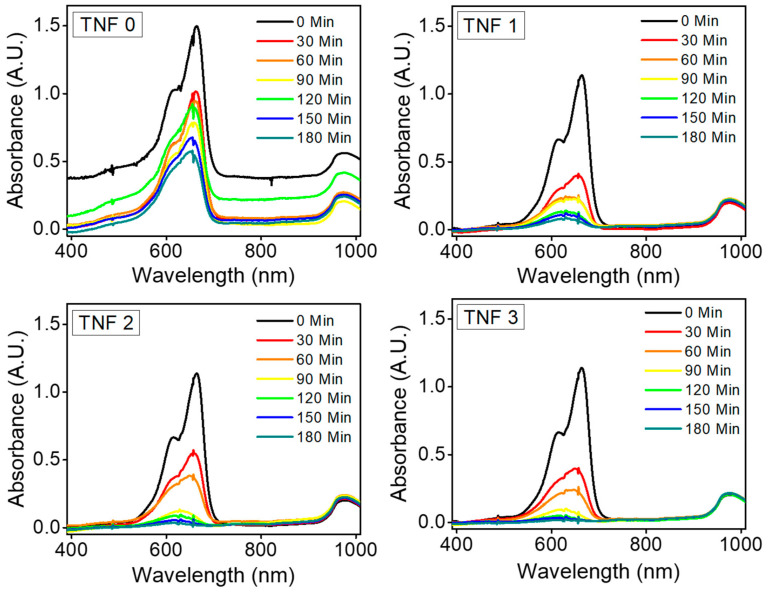
UV–Vis absorption spectra of photocatalytic degradation of methylene blue solution by TNFs.

**Figure 9 polymers-15-00134-f009:**
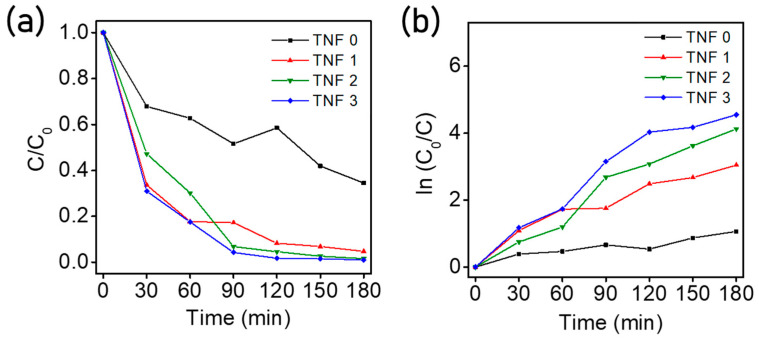
Photodegradation efficiency of methylene blue ((**a**): photodegradation rate of the TNF samples with UV–Vis light irradiation UV irradiation; (**b**): kinetic linear simulation curves of photodegradation under visible light for TNF samples).

**Figure 10 polymers-15-00134-f010:**
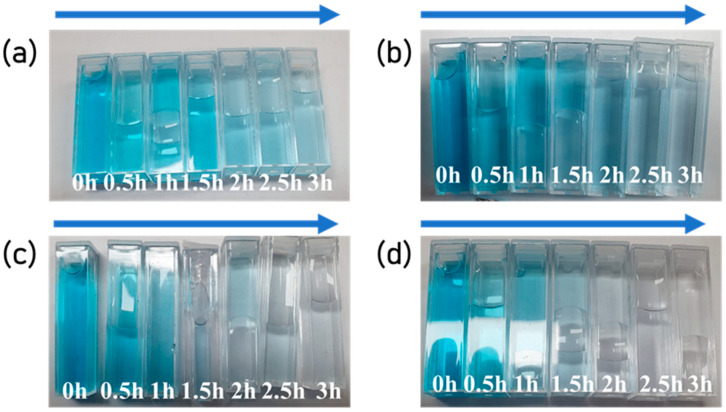
Photodegradation results of MB solution as time passed by TNF samples ((**a**): TNF 0; (**b**): TNF 1; (**c**): TNF 2; (**d**): TNF 3).

**Table 1 polymers-15-00134-t001:** Variables sample of precursors’ conditions (K-90, K-30).

	TNF 0	TNF 1	TNF 2	TNF 3
K-90: K-30 (g)	10 g: 0 g (11.76wt%)	5 g: 5 g (10.53wt%)	5 g: 5 g (10.53wt%)	5 g: 5 g (10.53wt%)
TTIP (g)	15 g (17.65wt%)	15 g (15.79wt%)	15 g (15.79wt%)	15 g (15.79wt%)
EtOH (g)	50 g (58.82wt%)	50 g (52.63wt%)	50 g (52.63wt%)	50 g (52.63wt%)
ACAC (g)	10 g (11.76wt%)	10 g (10.53wt%)	10 g (10.53wt%)	10 g (10.53wt%)
DIPA (g)	0 g (0wt%)	10 g (10.53wt%))	10 g (10.53wt%)	10 g (10.53wt%)
Heating rate	5 °C/min	5 °C/min	3 °C/min	1 °C/min

**Table 2 polymers-15-00134-t002:** BET test of electrospun TNF samples.

Sample Name	Specific Surface Area (m^2^/g)	Pore Volume (cm^3^/g)	BJH Desorption Average Pore Width (nm)	Micropore Area (m^2^/g)
TNF 0	17.2	0.05	13.37	11.4
TNF 1	23.8	0.09	15.64	14.4
TNF 2	27.4	0.06	8.7	14.0
TNF 3	23.5	0.09	16.2	17.4

## Data Availability

Not applicable.

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
