# Peer review of "Fabrications of Electrospun Mesoporous TiO_2_ Nanofibers with Various Amounts of PVP and Photocatalytic Properties on Methylene Blue (MB) Photodegradation"

_polymers, 2022, doi:10.3390/polym15010134_

Round 1
Reviewer 1 Report
The manuscript entitled “Fabrications of Electrospun Mesoporous TiO2 Nanofibers with Various PVP Content and Photocatalytic Properties on Methylene Blue (MB) Photodegradation”. This reviewer would like to consider this work for publication in the Polymers after addressing some concerns listed below.
Comments
The author should provide some quantitative information in the abstract section.
Materials and Fabrication of TiO2 nanofibers should be provide separately.
There is no citation. The author should cite more references in the results and discussion section.
The author should perform the FTIR spectra of various TiO2 nanofiber samples.
Figure 8. Photodegradation results of MB solution as passed time by TNF samples (a: TNF 0; b: TNF 1; c: TNF 2; d: TNF 3). The author should mention the time in the insert photographs.
In conclusion section should revised with outstanding point of this work.
Reviewer 2 Report
Comments: This work reported a mesoporous TiO2 nanofibers for photo catalyzing Methylene Blue (MB). There are some questions that should be further explained and some experiments need to be supplemented.
1) In TiO2 nanofiber preparation process, how to determine the calcination temperature? Whether the TiO2 nanofiber contains impurities? TG analysis should be added in the paper.
2) In the process of fiber preparation, the difference between K30 and K90 PVP used is mainly the molecular weight, so is it possible to achieve the same effect of controlling the surface properties by choosing one of them to control its quality?
3) It is mentioned that “compared to TNF 0, TNF 1,2 and 3 showed a mesoporous surface”. The mesoporous structure is usually confirmed by a BET method instead of SEM. Besides, in the pore properties characterization, TNF 0 is also a mesoporous structure, so the expression of the pore structure in this paper is not appropriate and needs to be revised.
4) In the last part of the introduction, it is proposed that “the mesoporous nanofibers showed improved photocatalytic efficiency than ordinary nanofibers”. Literature comparison is needed to better understanding the material’s performance.
5) Researches on photocatalytic nanofibers can be found elsewhere, it is essential to evaluate the stability and reuse ability of the material.
6) The heating rate has a negative effect on improving the specific surface area and the catalytic performance of the nanofiber. How to ensure TNF3 is the best one?

Reviewer 4 Report
The manuscript seems interesting due to the subject matter and the proposed solution. In fact, many strategies to improve the photocatalytic properties of titanium oxide are reported in the literature, also on nanofibers developed by electrospinning, but the present study would consist of a simple solution based essentially on both the molecular weight of the carrier polymer and on the annealing rate. However, many details as well as the results obtained remain dark and should be clarified.
A list of comments and suggestions for the Authors follows:
1) More details about the role and the effects of DIPA within the electrospun precursor solution and the resulting fibers should be explained (not only as a surfactant).
2) Has been TNF0 treated with different thermal rates as for TNF 1,2 and 3? If NO, the Authors should justify it!
3) The Authors should be spend more words to describe the resulting morphology of nanofibers differently designed and treated. For instance, TNF0 NFs look more homogeneous, despite of TNF2, as well as TNF3 NFs appear fragmented. Such results should be explained! Moreover, a higher molecular weight polymer tends to generate thicker fibers than a lower molecular weight polymer. In this case however, the mixture of high molecular weight PVP with one of low MW seems to create fibers with thicker diameters, decreasing only with the decreasing thermal rate. Therefore the Authors should comment more coherently on their findings!
4) The thermal rate affects the NFs crystallite shapes and diameters, but the Authors comments appear to contradict the results reported (r. 172-175). They should clarify them!
5) R.189-191: the same sentence is duplicated.
6) There are various punctuation errors!
7) Should the Authors describe the results obtained related to the measurements of the pores and the exposed surface area according to a more intelligible description? Too many concepts and parameters described all together cause confusion in the reader.
Round 2
Reviewer 1 Report
The author gave a detailed and reasonable response to the comments of the reviewers. I recommend it to be accepted.
Reviewer 3 Report
Dear Author,
all the points have been clarified.
Reviewer 4 Report
The manuscript has been revised according to the Reviewer's comments, thus it looks clearer and more useful to the scientific community.
Therefore I suggest to consider the revised manuscript suitable for publication.